# A Novel Antimicrobial Peptide Sparanegtin Identified in *Scylla paramamosain* Showing Antimicrobial Activity and Immunoprotective Role In Vitro and Vivo

**DOI:** 10.3390/ijms23010015

**Published:** 2021-12-21

**Authors:** Xuewu Zhu, Fangyi Chen, Shuang Li, Hui Peng, Ke-Jian Wang

**Affiliations:** 1State Key Laboratory of Marine Environmental Science, College of Ocean & Earth Sciences, Xiamen University, Xiamen 361005, China; zhuxuewu@stu.xmu.edu.cn (X.Z.); chenfangyi@xmu.edu.cn (F.C.); 22320170154929@stu.xmu.edu.cn (S.L.); penghui@xmu.edu.cn (H.P.); 2State-Province Joint Engineering Laboratory of Marine Bioproducts and Technology, College of Ocean & Earth Sciences, Xiamen University, Xiamen 361005, China; 3Fujian Innovation Research Institute for Marine Biological Antimicrobial Peptide Industrial Technology, College of Ocean & Earth Sciences, Xiamen University, Xiamen 361005, China

**Keywords:** *Scylla paramamosain*, antimicrobial peptide, Sparanegtin, antimicrobial activity, immunoprotective role

## Abstract

The abuse of antibiotics in aquaculture and livestock no doubt has exacerbated the increase in antibiotic-resistant bacteria, which imposes serious threats to animal and human health. The exploration of substitutes for antibiotics from marine animals has become a promising area of research, and antimicrobial peptides (AMPs) are worth investigating and considering as potential alternatives to antibiotics. In the study, we identified a novel AMP gene from the mud crab *Scylla paramamosain* and named it Sparanegtin. *Sparanegtin* transcripts were most abundant in the testis of male crabs and significantly expressed with the challenge of lipopolysaccharide (LPS) or *Vibrio alginolyticus*. The recombinant Sparanegtin (rSparanegtin) was expressed in *Escherichia coli* and purified. rSparanegtin exhibited activity against Gram-positive and Gram-negative bacteria and had potent binding affinity with several polysaccharides. In addition, rSparanegtin exerted damaging activity on the cell walls and surfaces of *P. aeruginosa* with rougher and fragmented appearance. Interestingly, although rSparanegtin did not show activity against *V. alginolyticus* in vitro, it played an immunoprotective role in *S. paramamosain* and exerted an immunomodulatory effect by modulating several immune-related genes against *V. alginolyticus* infection through significantly reducing the bacterial load in the gills and hepatopancreas and increasing the survival rate of crabs.

## 1. Introduction

It is estimated that China accounts for over 60% of the global aquaculture production under the accelerated development of aquaculture industry [1]. Accordingly, various diseases often occur in the process of aquaculture, especially the bacterial infectious diseases, which cause the antibiotics widely used in aquaculture, either as pharmaceuticals in control diseases or routinely used in feedstuff as additives. The abuse of antibiotics leads to antibiotic residual problems in aquatic products. Through the consumption of aquatic products tainted by antibiotics, humans may acquire adverse drug reactions [2]. In particular, the abuse of antibiotics increased numbers of antibiotic-resistant pathogenic microorganisms in the aquatic environment, which poses a challenge to the development and use of antibiotic strategies to control fish diseases [3,4]. As it is known that antibiotic medications have been widely used not only in clinical treatment and the prevention of microbial infections, but also in feedstuffs [5], the wide spread of antimicrobial resistance (AMR) seriously affects animal and human health [6]. To control the antibiotic-resistant pathogens, a variety of effective first-line drug treatments (such as chloramphenicol, erythromycin, and terramycin) have recently been developed to control aquatic bacteria; however, these drugs often negatively affect many organisms, including fish and humans [3,7]. Therefore, the exploration and development of effective alternatives to substitute for antibiotics becomes a promising research hotspot.

It is well known that marine invertebrates including crustaceans mainly depend on innate immune defense to protect themselves against invading pathogens. Of various effective immune-related components, the antimicrobial peptides (AMPs) are the most concerned because they play a significant role in innate immunity and serve as effective defense weapons against bacterial, fungal, and viral infections [8,9]. The antimicrobial mechanism of most AMPs is to disrupt the membrane integrity of invading microorganisms [10,11]. Compared with antibiotics, AMPs can offer multiple advantages as candidates for the development of antimicrobial agents, as their uses may include acting alone or in synergy with other antimicrobial agents to reduce the effective bactericidal concentration and thereby reduce cytotoxicity, and it is not easy to induce drug resistance in bacteria [12]. In addition to direct antibacterial functions, AMPs have an important capability to regulate the innate immune system [13]. Therefore, AMPs can not only improve the immune resistance of aquatic animals but also alleviate the problems of bacterial resistance and antibiotic contamination of aquatic products in aquaculture.

AMPs can also produce immunological protection against bacterial challenge in vivo. Epinecidin-1, a synthetic 21-mer antimicrobial peptide originally identified from grouper (*Epinephelus coioides*), significantly improves the survival rate of zebrafish infected with *Vibrio vulnificus* [14]. LcLEAP-2C from large yellow croaker (*Larimichthys crocea*) can reduce the mortality of large yellow croaker after *V. alginolyticus* challenge [15], and white spot syndrome virus (WSSV) pre-incubated with anti-lipopolysaccharide factor (ALF) results in an increased survival rate of red claw crayfish (*Cherax quadricarinatus*) [16]. Similarly, in our laboratory, the recombinant product of one AMP SpHyastatin, which is identified in *S. paramamosain* can enhance the protection of the host against *Vibrio parahaemolyticus* infection in crabs [17]; there are two other AMPs: rSpALF7 could obviously improve the survival of crabs infected by *V. alginolyticus* [18] and rScyreprocin significantly decreased the mortality of *Vibrio harveyi*-infected marine medaka [19]. The action mechanism of antimicrobial peptides in vivo has been also investigated. Several recent studies have found that the administration of AMPs to fish can lead to a decrease in the number of bacteria in tissues, showing a direct antibacterial activity in vivo [20]. Some AMPs could be attributed to their ability to enhance immune response by modulating host gene expression [13], inducing or inhibiting cytokine production [20], and promoting the production of antimicrobial substances such as lysozymes and antioxidant enzymes [21].

In the study, based on the transcriptome database of *S. paramamosain* established by our laboratory, we identified an uncharacterized gene for the first time and named it Sparanegtin. The expression profiles of *Sparanegtin* in *S. paramamosain* with the challenge of LPS or *V. alginolyticus* were investigated. The recombinant product of Sparanegtin (rSparanegtin) in a prokaryotic expression system *Escherichia coli* was obtained. The antimicrobial activity assay, scanning electron microscopy (SEM), observation, and microbial surface components binding assays were performed to analyze the antimicrobial features of rSparanegtin against various microorganisms in vitro. In addition, the effect of rSparanegtin in vivo was evaluated by detecting the bacterial clearance ability in the gills and hepatopancreas of *S. paramamosain* infected with *V. alginolyticus*, as well as any effect on the expression patterns of some immune-related genes after the in vivo administration of rSparanegtin. This study aims to preliminarily study the function, immune-protective effect, and related mechanism of Sparanegtin, providing effective strategies for mud crab aquaculture disease control. This study aims to characterize the new AMP Sparanegtin, elucidating its immune-protective effect and the underlying mechanism and thus developing a potential effective antimicrobial agent that could be substituted for antibiotics to be used in animal husbandry or medicine in the future.

## 2. Results

### 2.1. Cloning and Sequence Analysis of Sparanegtin

The full-length cDNA sequence of *Sparanegtin* was obtained, which is 525 bp, including a 252-bp open reading frame (GenBank accession number: MN612064). It had a predicted signal peptide of 23aa, and the cleavage position is between Gly-23 and Ala-24. The mature peptide contained 60 amino acid residues, and its calculated mass is 5.818 kDa with an estimated isoelectric point (pI) of 5.2, the total net charge of -1 (Figure 1A). The predicted tertiary structure of Sparanegtin contains three α-helices (Figure 1B).

### 2.2. Gene Expression Profiles of Sparanegtin

The qPCR results showed that *Sparanegtin* was widely distributed in different tissues (Figure 2A,B). In male adult crabs, *Sparanegtin* was dominantly expressed in the testis (Figure 2A), and the highest expression level of *Sparanegtin* was found in the hemocytes of female adult crabs (Figure 2B). We further investigated the expression profiles of *Sparanegtin* in the testis and hemocytes of male crabs after LPS or *V. alginolyticus* challenge (Figure 2C–F). In the testis, the expression of *Sparanegtin* was significantly down-regulated by LPS challenge at 3 hpi (Figure 2C), while it showed significant up-regulation at 3 hpi and 72 hpi under *V. alginolyticus* challenge (Figure 2D). In the hemocytes, *Sparanegtin* gene was significantly up-regulated at 3 hpi under both LPS and bacterial challenge (Figure 2E,F).

### 2.3. rSparanegtin Shows Antimicrobial Activity

The recombinant product of Sparanegtin (rSparanegtin) was successfully expressed in *E. coli*. SDS-PAGE analysis showed that the purity of rTrx and rSparanegtin was high, as shown in Figure 3A. In addition, the results from the mass spectrometry also confirmed that the purified protein was the target protein rSparanegtin (Appendix A). The antimicrobial activity of rSparanegtin was determined. As shown in Table 1, rSparanegtin displayed good antimicrobial activities against several Gram-negative (*E. coli*, *P. aeruginosa*, *P. stutzeri*, *P. fluorescens*, *S. flexneri*), Gram-positive (*B. subtilis*, *C. glutamicum*, *S. aureus*) bacteria (MICs ranging from 12 to 48 μM and MBCs ranging from 24 to 48 μM), and yeast (*C. neoformans* and *P. pastoris* GS115) (MICs ranging from 24 to 48 µM).

### 2.4. Preliminary Study on the Antibacterial Mechanism of rSparanegtin

#### 2.4.1. Binding Properties

ELISA assay was used to investigate the binding properties of rSparanegtin to different microbial surface molecules and bacteria. In order to evaluate whether the label protein Trx would have any effect on the following results, rTrx was selected as the control group. Compared with the rTrx group, rSparanegtin had strong binding affinity with LPS, LTA, and PGN in a concentration-dependent manner, and their calculated apparent dissociation constants (K_d_) were 0.2375, 0.3905, and 0.6246 μM, respectively (Figure 3B). 

#### 2.4.2. Killing Kinetic

The results of the time-killing kinetic assay were applied to further evaluate the bactericidal activity of rSparanegtin. When rSparanegtin was incubated with *P. aeruginosa* at a concentration of 48 µM, all bacteria could be killed after 4 h of incubation (Figure 3C). 

#### 2.4.3. rSparanegtin Induces Morphological Changes in Microorganisms

In order to study the antibacterial mechanism of rSparanegtin against *P. aeruginosa*, SEM was employed to observe the morphological changes of the microbial membrane after rSparanegtin and rTrx treatment. After incubating with rSparanegtin and rTrx for a certain period of time, the SEM images of *P. aeruginosa* showed a significant destruction of membrane integrity and even leakage of cell contents compared with the control group and rTrx group (Figure 3D).

### 2.5. rSparanegtin Shows No Cytotoxicity and Could Reduce the V. alginolyticus Endotoxin Level In Vitro

The cytotoxicity of rSparanegtin was analyzed using primarily cultured crab hemocytes, HEK-293T and NCI-H460. As shown in Figure 4A–C, rSparanegtin showed no cytotoxicity. 

In addition, it was found that rSparanegtin treatment could significantly reduce the endotoxin level of *V. alginolyticus*, which also showed a dose-dependent manner. Under the treatment of 48 μM, the endotoxin level was reduced by about 70% (Figure 4D).

### 2.6. The Immunoprotective Effect of rSparanegtin on S. paramamosain

#### 2.6.1. Survival Rate Comparison

To investigate the in vivo protective effect of rSparanegtin, male mud crabs were challenged with different groups, including the PBS and *V. alginolyticus* pre-incubation group (short as PBS group), rTrx and *V. alginolyticus* pre-incubation group (short as rTrx group), and rSparanegtin and *V. alginolyticus* pre-incubation group (short as rSparanegtin group). As shown in Figure 4E, 48 h after different treatments, the survival rate of the crab PBS and rTrx groups dropped to 50%, while the survival rate of the rSparanegtin group was around 75%. Crabs in the PBS and rTrx groups died faster, and none of them survived 120 h after injection, while the survival rate of the rSparanegtin group was still about 40% (*p* < 0.05) (Figure 4E). 

#### 2.6.2. Pre-Incubation of rSparanegtin and *V. alginolyticus* Reduces Bacterial Load in the Tissues 

Bacterial clearance represents a major endpoint of innate host immunity in response to infection. As we all know, AMPs are important components of the innate immune system. We evaluated the ability of rSparanegtin to eliminate bacteria in the tissues of mud crabs under different treatments as mentioned above. As shown in Figure 5A, compared with the PBS and rTrx groups, the rSparanegtin group showed a significant reduction in *V. alginolyticus* load in the gills at the 3, 6, 12, and 24 hpi (Figure 5A). In the hepatopancreas, the *V. alginolyticus* load significantly decreased at 6, 12, and 24 hpi (Figure 5B).

#### 2.6.3. Pre-Incubation of rSparanegtin and *V. alginolyticus* Modulate Immune-Related Gene Expression Profiles 

The results of qPCR showed the effect of pre-incubation of rSparanegtin and *V. alginolyticus* on the immune response of *S. paramomosain* (Figure 6). Compared with the PBS group and the rTrx group, the transcription levels of the canonical components of the immune pathway (including *Sp*Toll2, *Sp*Myd88, and *Sp*STAT), two AMPs (SpHyastatin and *Sp*ALF2), and antioxidant enzyme genes (including *Sp*CAT, *Sp*SOD, and *Sp*GPx) were increased significantly at 6 h in the rSparanegtin group. 

## 3. Discussion

In the study, based on the transcriptome database of *S. paramamosain* established by our laboratory, we identified a novel AMP and named it Sparanegtin. According to the theoretical pI 5.2 of its mature peptide, Sparanegtin is an anionic AMP. As it is known, most reported AMPs are cationic peptides; however, more anionic AMPs have been gradually identified in different species in recent years and also have a potent antimicrobial activity. Dermcidin is a novel human antibiotic peptide secreted by sweat glands and has a net negative charge of –5 that shows antimicrobial activity in response to a variety of pathogenic microorganisms [22]. The three antifungal peptides from the *Litopenaeus stylirostris* and *Litopenaeus vannamei* have a negative net charge at physiological pH with a pI and a broad spectrum of antifungal activity [23]. Our previous studies report that two novel AMPs, Scygonadin [24] and its homologous SCY2 [25], are anionic peptides and both have antimicrobial activity. In the present study, it was found that rSparanegtin displayed a potent activity against several Gram-negative bacteria (*E. coli*, *P. aeruginosa*, *P. stutzeri*, *P. fluorescens*, and *S. flexneri*) (MICs ranging from 12 to 48 μM), Gram-positive bacteria (*B. subtilis*, *C. glutamicum*, and *S. aureus*), and yeast (*C. neoformans* and *P. pastoris GS115*) (MICs ranging from 24 to 48 μM) (Table 1). The in vivo expression pattern of the Sparanegtin gene was tissue-specific. The mRNA transcripts of Sparanegtin were highly expressed in the testis of male crabs. In addition, some known AMPs are sex-specifically expressed; for instance, Adropin is specifically expressed in the ejaculatory duct of *Drosophila melanogaster* [26], as observed in our early study on Scygonadin that is dominantly expressed in the ejaculatory duct of male mud crabs and is involved in the reproductive immunity [24]. A recently reported AMP, scyreprocin, is identified as an interacting partner of SCY2 from the reproductive system of male *S. paramamosain* and highly expressed in the testis [19]. It is known that testes are organs of the male reproductive system of decapod crustaceans and harbor germ cells and produces spermatozoa [27], as well as being functional either at the beginning or during the entire spermatogenesis process [28]. Therefore, Sparanegtin that is highly present in testes may play an immune defense role in spermatogenesis and the reproduction process of male crabs.

Binding to the surface of microorganisms is the first step for AMP to exert its antimicrobial effect. In order to better understand the underlying antimicrobial mechanism of AMPs, microbial cell wall polysaccharides binding assays were conducted in this study. The present study revealed that rSparanegtin had a strong binding ability to LPS, PGN, and LTA in a concentration-dependent manner and exhibited a higher binding ability to LPS than to PGN and LTA. Many AMPs are reported with similar activities via binding to microbial cell wall polysaccharides. rPcALF1 from red swamp crayfish (*Procambarus clarkii*) could bind with different amounts of microbial polysaccharides, mostly with LPS, followed by glucan, and the least with LTA, and then find that it has stronger antibacterial activity against Gram-negative bacteria [29]. In *Marsupenaeus japonicus*, MjCru I-1 could agglutinate bacteria and bind to bacteria by binding to the bacterial cell wall molecules including LPS, LTA, and PGN. MjCru I-1 had antibacterial activity against some bacteria by destroying the membrane of bacteria [30]. rLvCrustinB from Pacific white shrimp *Litopenaeus vannamei* directly binds to polysaccharides, including PGN, LTA, and LPS, indicating that LvCrustinB may be involved in the defense against Gram-positive and Gram-negative bacteria [31]. In the study, the SEM images of *P. aeruginosa* showed a significantly destruction of membrane integrity and even leakage of cell contents, suggesting that the activities of rSparanegtin may be via the interaction with the specific components of bacterial cell wall. This is consistent with the fact that rSparanegtin has high antimicrobial activity against *P. aeruginosa*. The antimicrobial mechanism of rSparanegtin may be similar to that of most AMPs that destroy the integrity of the microbial membrane, which leads to the leakage of the cytoplasmic contents and ultimately kills them [32].

It was interesting to note in the study that the survival rate of *S. paramamosain* challenged with *V. alginolyticus* was increased when rSparanegtin was given to crabs; correspondingly, there was a significant reduction in *V. alginolyticus* load in the gills and hepatopancreas at 6, 12, and 24 h. The results suggested that rSparanegtin might exert an immunological defense against the invading *V. alginolyticus* by which the survival rate of crabs was enhanced. Analysis of the *Sparanegtin* gene in vivo demonstrated that this peptide was significantly expressed in the testis at 3 h and 72 h or hemocytes at 3 h with the *V. alginolyticus* challenge; meanwhile, other AMPs such as SpHyastatin and *Sp*ALF2 as well as signal pathway associated genes such as *Sp*Toll2, *Sp*Myd88, and *Sp*STAT were up-regulated at 6 h. All of these findings suggested that Sparanegtin may directly participate in the immune response or indirectly play a role by inducing the expression of other immune-associated genes with the injection of rSparanegtin when bacterial infection occurred in crabs; that means that Sparanegtin may generate immunoprotective and immunomodulatory activities. AMPs as products of immune response are testified to play important roles in killing or cleaning the infected pathogens directly. The significant expression of SpHyastatin and *Sp*ALF2 at 6 h after *V. alginolyticus* challenge might be due to the immunomodulatory effect of rSparanegtin. In a previous study on SpHyastatin, this peptide is down-regulated at 24 h and then up-regulated at 96 h but does not show any change in expression at 6 h after bacterial challenge [33], suggesting that the expression of SpHyastatin might be directly induced by the injection of rSparanegtin. The significant expression of *Sp*Toll2, *Sp*Myd88, and *Sp*STAT implied that the immune-associated signal pathways participated in the defense against the *V. alginolyticus* challenge and may induce the activation of downstream effectors such as AMPs. 

In addition to resistance to a variety of pathogenic microorganisms, AMPs are also reported to regulate the expression of other immune genes [34]. We found that the pre-incubation of rSparanegtin and *V. alginolyticus* could induce the transcription levels of several immune-related genes, including immune signaling pathway-related genes (*Sp*Toll2, *Sp*Myd88, *Sp*STAT), AMPs (SpHyastatin and *Sp*ALF2), and antioxidant-associated genes (*Sp*SOD, *Sp*CAT, and *Sp*GPx). Such immunoenhancing properties are demonstrated in several other marine-derived AMPs, for example, shrimp and *limulus* anti-lipopolysaccharide factor [35,36,37]. The innate humoral immune response is mainly mediated by three immune signaling pathways, namely, the Toll pathway, IMD pathway, and JAK/STAT pathway [38]. By regulating or stimulating the Toll signaling pathway, the production of some immune factors related to its downstream pathway, such as antimicrobial peptides (AMPs), can be activated against microbial infection [39]. The JAK/STAT signaling pathway positively regulates AMP gene expression that plays an important role in immune response [40]. In this study, the expression trend of both AMPs (SpHyastatin and *Sp*ALF2) genes was consistent with the expression of *Sp*Toll2, *Sp*MyD88, and *Sp*STAT, suggesting that the expression of both AMPs might be regulated through the Toll and JAK/STAT pathways. The up-regulation of SpHyastatin and *Sp*ALF2 may participate in eliminating the infected bacteria. In addition, bacterial infection can prompt the body to produce ROS, and excessive ROS will cause tissue damage and inflammation [41,42]. The up-regulated expression of antioxidant enzymes (*Sp*SOD, *Sp*CAT, and *Sp*GPx) might be associated with the action of removing ROS in vivo. These results suggested that Sparanegtin was likely generating an immunomodulatory effect that helps eliminate the invading bacteria. 

The interactions among the induced expression of AMPs, clear degree of infected bacterial numbers, and survival rate of marine animals have been much reported in previous studies. For example, MjALF-E2 were upregulated by bacterial challenge and could promote the clearance of bacteria in vivo. After knockdown of MjALF-E2 and infection with *Vibrio anguillarum*, shrimp showed high and rapid mortality compared with GFPi shrimp, suggesting that MjALF-E2 serves a protective function against bacterial infection in shrimp [43]. A crustin gene PcCru isolated from red swamp crayfish *Procambarus clarkia* is significantly induced by bacterial stimulations at both the translational and transcriptional levels and could protect crayfish from infection by the pathogenic bacteria *Aeromonas hydrophila* in vivo [44]. In a bacteria challenge test, As-CATH4 and 5 (two vertebrate-derived cathelicidins family HDPs) could significantly decrease the bacterial numbers in crabs and increase the survival rates of crabs in both pre-stimulation and co-stimulation groups [45]. Similarly, the expression level of PcALF1 is induced by bacteria, and the injection of PcALF1 in crayfish (*Procambarus clarkii*) enhances the elimination of bacteria in vivo [29]. Our previous studies on other two AMPs, SCY2 and SpHyastatin, also show an immunoprotective effect on *S. paramamosain*, although both have differential antimicrobial activity and in vivo expression patterns. rSpHyastatin, a peptide that is highly expressed in hemolymphs with bacterial challenge, could confer immune-protective resistance against pathogenic challenge in *S. paramamosain*, causing less significant change in level of the mRNA expression of all tested immune and antioxidant-associated genes [17]. For SCY2, even though its gene expression is uniquely expressed during the mating of crabs and could not be directly induced by the injection of bacteria, rSCY2 could significantly increase the survival rate of *S. paramamosain* [46]. It is worth noting that rSparanegtin had no inhibitory or killing effect on cultured *V. alginolyticus* in vitro; however, it could be significantly expressed at some timepoints with the *V. alginolyticus* challenge in vivo and could significantly improve the survival rate of *S. paramamosain* after *V. alginolyticus* challenge as well as reduce *V. alginolyticus* load in the gills and hepatopancreas. The similar phenomenon is also found in our early study on an AMP SpHyastatin that is also identified in *S. paramamosain* [17].

## 4. Materials and Methods

### 4.1. Microorganism Strains 

All strains were purchased from the China General Microbiological Culture Collection Center (CGMCC), including *Staphylococcus aureus* (CGMCC No. 1.2465), *Staphylococcus epidermidis* (CGMCC No. 1.4260), *Escherichia coli* (CGMCC No. 1.2389), *Pseudomonas aeruginosa* (CGMCC No. 1.2421), *Pseudomonas fluorescens* (CGMCC No. 1.1802), *Shigella flexneri* (CGMCC No. 1.1868), *Bacillus subtilis* (CGMCC No. 1.3358), *Cryptococcus neoformans* (CGMCC No. 2.1563), and *Vibrio alginolyticus* (CGMCC No. 1.1833). *Pichia pastoris* (GS115) was purchased from Invitrogen (Thermo Fisher Scientific, Waltham, MA, USA). 

### 4.2. Animals, Challenge and Tissue Collection

Mud crabs (*S. paramamosain*) were purchased from the Zhangzhou Crab Farm (Fujian, China). Healthy male and female adult mud crabs (body weight 300 ± 30 g, *n* = 5) were dissected, and the tissues including testis, anterior vas deferens, seminal vesicle, posterior vas deferens, ejaculatory duct, posterior ejaculatory duct, penis, ovary, spermathecae, reproductive duct, muscle, thoracic ganglion, gills, brain, midgut, subcuticular epidermis, eye stalk, heart, hepatopancreas, and stomach were collected. Hemocytes were isolated from the hemolymph as described previously [47]. For the challenge experiment, adult male crabs (body weight 300 ± 30 g, *n* = 5) were injected with LPS at a dosage of 0.5 mg kg^−1^ or *V. alginolyticus* (1 × 10^6^ CFU crab^−1^). Crabs injected with crab saline (NaCl, 496 mM; KCl, 9.52 mM; MgSO_4_, 12.8 mM; CaCl_2_, 16.2 mM; MgCl_2_, 0.84 mM; NaHCO_3_, 5.95 mM; HEPES, 20 mM; pH 7.4) were set up as the control group. Tissue samples (testes and hemocytes) were collected at 3, 6, 12, 24, 48, and 72 h post-injection (hpi). All tissues were stored at −80 °C until use. All animal procedures were carried out in strict accordance with the National Institute of Health Guidelines for the Care and Use of Laboratory Animals and were approved by the Animal Welfare and Ethics Committee of Xiamen University. 

### 4.3. Cloning, Expression, Purification, and Analysis of Recombinant Proteins 

Total RNA of testis was extracted using TRIzol™ reagent (Invitrogen, Carlsbad, CA, USA) and cDNA was generated using PrimeScript™ RT reagent Kit with gDNA Eraser Kit (Takara, China). The cDNA templates for 5′- and 3′-random amplification of cDNA ends (RACE) PCR were synthesized using a SMARTer^®^ RACE 5′/3′ Kit (Takara, Dalian, China). Gene-specific primers were designed based on the partial sequences obtained from the transcriptome database established by our laboratory (Table 2). The amplified fragments were cloned into the pMD18-T vector (Takara, Dalian, China) and sequenced by Borui Biotechnology Ltd. (Xiamen, China). 

The open reading frame of Sparanegtin was constructed into the pET-32a (+) vector (with 6× His tag and thioredoxin (Trx) tag) and transformed into *E. coli* BL21 (DE3) and further expressed (the specific primer sequences were listed in Table 2). A pET32a (+) vector with only 6× His tag and Trx (thioredoxin) tag was constructed, and the expressed product was used as a control. Isopropyl β-d-Thiogalactoside (IPTG) was added to a final concentration of 0.5 mM to induce protein expression at 28 °C for 8 h. The recombinant Sparanegtin (rSparanegtin) was expressed and purified through HisTrap TM FF crude (GE Healthcare, Chicago, IL, USA) on the ÄKTA Pure system (GE Healthcare, Chicago, IL, USA) according to the standard protocol. The purified proteins were dialyzed and concentrated, and the protein concentration was determined by Bradford assay. The purified proteins were confirmed by sodium dodecyl sulfate-polyacrylamide gel electrophoresis (SDS-PAGE), Western blotting, and mass spectrometry identification. All recombinant proteins were stored at −80 °C

### 4.4. Quantitative Real-Time PCR

The total RNA of all samples was extracted and cDNA was synthesized as described above. Quantitative reverse transcription PCR (qRT-PCR) was performed using the cDNA as the template to detect the expression level of Sparanegtin in a real-time thermal cycler (ABI 7500, Waltham, MA, USA) using FastStart DNA Master SYBR Green I (Roche Diagnostics, Mannheim, Germany). The expression profiles of Sparanegtin gene in various adult crab tissues were determined by absolute quantitative real-time PCR (qPCR) and the expression changes of Sparanegtin during the response patterns of Sparanegtin gene to LPS and *V. alginolyticus* challenge were analyzed by relative qPCR. The specific primer sequences (Sparanegtin-qPCR-F/Sparanegtin-qPCR-R, GADPH-qPCR-F/GADPH-qPCR-R) are listed in Table 1. The qPCR cycle conditions were set as follows: an initial denaturing step at 95 °C for 10 min, 40 cycles at 95 °C for 15 s, 60 °C for 30 s, and 72 °C for 1 min. The 2^−ΔΔCt^ algorithm was applied to the expression profile analysis [48].

### 4.5. Antimicrobial Assay

Microorganisms in the logarithmic growth phase were harvested and used to evaluate the antimicrobial activity of rSparanegtin. The minimum inhibitory concentration (MIC) and the minimum bactericidal concentration (MBC) were determined according to the previously described liquid growth inhibition assay, which were performed three times independently [49]. Compared with the negative control, the MIC value is defined as the lowest protein concentration that does not induce visible bacterial growth. Then, we spread the culture without visible bacterial growth on a solid medium plate. The MBC is the concentration that kills more than 99.9% of the microorganisms after incubation at 28 or 37 °C for 24 h.

### 4.6. Binding Assays

In order to determine the binding properties of rSparanegtin with lipopolysaccharides (LPS B5, Sigma, St. Louis, MO, USA), lipoteichoic acid (LTA, L2515, Sigma, St. Louis, MO, USA), and peptidoglycan (PGN from *Bacillus subtilis*, Sigma, St. Louis, MO, USA), a modified ELISA assay was performed as described previously [19]. Briefly, a 96-well ELISA plate was coated overnight with LPS, LTA, and PGN at 4 °C; then, it was blocked with 5% skimmed milk and incubated with a serial dilution of rSparanegtin and rTrx (0 to 5 μg mL^−^^1^) for 2 h at 37 °C. Bound peptides were detected by incubation with mouse anti-His antibody (1:3000, prepared in 1% skimmed milk) followed by adding goat anti-mouse HRP antibody (1:5000, prepared in 1% skim milk). After the colorimetric reaction, the absorbance at 450 nm was measured using a microplate reader (TECAN GENios, GMI, Brooklyn Park, MN, USA). The independent assays were performed three times. The binding parameters, apparent dissociation constant (Kd), and maximum binding (Amax) were determined using non-linear fitting as A = Amax [L] / (Kd + [L]), where A is the absorbance at 450 nm and [L] is the protein concentration [17].

### 4.7. Time-Killing Kinetic Assay

The Gram-negative bacteria *P. aeruginosa* were subjected for time-killing kinetic assay according to the previous description. rSparanegtin was incubated with bacteria at a concentration of 48 µM. The cultures were sampled and plated at different time points (*n* = 3). The plates were incubated at 37 °C for 24 h, and the total viable count (TVC) was determined. The independent experiments were performed three times.

### 4.8. SEM Observation

SEM was used to further study the antibacterial mechanism of rSparanegtin. *P. aeruginosa* (5 × 10^7^ CFU mL^−1^) was prepared as described in the antimicrobial assay. PBS, rTrx, and rSparanegtin were separately added into each individual culture medium and incubated at a concentration of 48 µM for 30 min. The microbial cells were collected and fixed with pre-cooled 2.5% glutaraldehyde at 4 °C for 2 h. Then, the samples were dehydrated with a graded series of ethanol (30%, 50%, 70%, 80%, 95%, and 100%) and further dehydrated in a critical point dryer (EM CPD300, Leica, Wetzlar, Germany) and gold coated [50]. Finally, the change in morphology of the bacteria was observed by SEM (SUPRA 55 SAPPHIRE, Carl Zeiss, Oberkochen, Germany).

### 4.9. Cytotoxicity Assay

The cytotoxicity of rSparanegtin was evaluated using hemocytes from *S. paramamosain*. The hemocytes of *S. paramamosain* were isolated as previously described [51]. Briefly, the hemocytes were maintained in L-15 medium prepared in crab saline and supplemented with 5% fetal bovine serum, inoculated on a 96-well cell culture plate with approximately 10^4^ cells well^−1^, and incubated overnight at 26 °C. HEK-293T cells were maintained in Dulbecco’s Modified Eagle Medium supplemented with 10% fetal bovine serum, and NCI-H460 cells were maintained in Roswell Park Memorial Institute 1640 supplemented with 10% fetal bovine serum. HEK-293T and NCI-H460 cells were inoculated on a 96-well cell culture plate and incubated at 37 °C with 5% CO_2_ overnight. Finally, all the cells were incubated with culture medium supplemented with various concentrations of rSparanegtin (3, 6, 12, 24 and 48 μM, *n* = 3). After 24 h of incubation, cell viability was assessed using a CellTiter 96 R ^®^ Aqueous Kit (Promega, Madison, WI, USA). The independent experiments were carried out three times. 

### 4.10. Endotoxin Assay

The endotoxin level of *V. alginolyticus* after rSparanegtin treatment was detected by the Toxin Sensor™ Chromogenic LAL Endotoxin Assay Kit (GenScript, Piscataway, NJ, USA) following the manufacturer′s instructions [52]. When *V. alginolyticus* reached the logarithmic growth phase, they were collected and adjusted to a concentration of 10^7^ CFU/mL. Then, they were incubated with different concentrations of rSparanegtin (0, 12, 24, 48 µM, *n* = 3) at room temperature for 1 h and analyzed by a spectrophotometer at an absorbance of 545 nm (Agilent Technologies, Bayan Lepas, Malaysia). Each sample had three biological parallels. The independent experiments were carried out three times.

### 4.11. Evaluation of the In Vivo Activity of rSparanegtin on S. paramamosain Infected with V. alginolyticus

In order to investigate the in vivo protective effect of rSparanegtin, we performed a mortality comparison assay using male *S. paramamosain* (average weight 40 ± 5 g) infected with *V. alginolyticus*. rSparanegtin, rTrx, and *V. alginolyticus* were prepared in PBS. First, the recombinant protein (20 µg/crab) was incubated with *V. alginolyticus* (1 × 10^6^ CFU/crab) at room temperature for 1 h, and then, the mixture was injected into the base of the right fourth leg of crabs. The control group received an equal volume of *V. alginolyticus* diluted in PBS. Sixty crabs were divided into three groups (including PBS and *V. alginolyticus* pre-incubation group, rTrx and *V. alginolyticus* pre-incubation group, and rSparanegtin and *V. alginolyticus* pre-incubation group) with 20 crabs in each group. The survival rates of crabs in each group were recorded at different time points (3, 6, 9, 12, 24, 36, 48, 60, 72, 96, and 120 h).

### 4.12. Bacterial Load Assay and Quantification of Immune-Related Gene Expression after Different Treatment

In order to investigate the bacterial load in tissues, male *S. paramamosain* (average weight 40 ± 5 g each) was performed different treatments as described above. The crabs were dissected, and tissues including hemocytes, gills, and hepatopancreas were collected at different time points (3, 6, 12, and 24 h, *n* = 5). Gills and hepatopancreas (0.1–0.2 g fresh weight per tissue) were homogenized in PBS. Then, the tissue homogenates were spread on marine broth 2216E plates, and the plates were incubated at 28 °C for 24 h. The colonies were counted separately for each sample at each time point.

The total RNA of the collected tissues was extracted, and the cDNA was synthesized as described above. The expression profiles of several immune-related genes were analyzed by qRT-PCR. The GenBank accession numbers for those genes are listed as follows: *Sp*Toll2: SLM84439.1; *Sp*Myd88: KC342028.1; *Sp*STAT: KC711050.1; *Sp*SOD: FJ774661.1; *Sp*CAT: FJ774660.1; *Sp*GPx: JN565286.1; *Sp*ALF2: JQ069031.1 and SpHyastatin: AFY10070.1. The specific primers for these genes are summarized in Appendix A.

### 4.13. Statistical Analysis

The results are presented as the mean ± standard deviation (SD). For the absolute qPCR assays, statistical analyses were performed by one-way analysis of variance (ANOVA) following a Tukey post-test. For the relative qPCR assays, statistical analyses were performed by two-way ANOVA following a Bonferroni post-test. For cytotoxicity assays, statistical analysis was performed by one-way ANOVA following Dunnett’s post-test. For the mortality comparison assay, data were analyzed using the Kaplan–Meier log rank test. For the immune-related gene expression, one-way analysis of variance (ANOVA) was used for statistical analysis using SPSS 18.0 (IBM, Armonk, NY, USA) to determine the expression difference within groups. Significant levels were accepted at *p* < 0.05.

## 5. Conclusions

In summary, a new antimicrobial peptide named Sparanegtin was identified in *S. paramamosain*, and its transcripts were specifically distributed in tissues and significantly expressed with bacterial challenge. rSparanegtin had antimicrobial activity, and the antimicrobial mechanism involved initial damage to the outer membrane of bacteria, eventually resulting in the loss of cellular components and the complete collapse of the cell architecture. rSparanegtin showed no cytotoxicity and could reduce the *V. alginolyticus* endotoxin level in vitro. This AMP had an in vivo protective and immunomodulatory effect in *S. paramamosain* that could reduce the bacterial load in tissues and enhance the survival rate of crabs challenged with *V. alginolyticus*. Taken together, Sparanegtin might be a potential effective antimicrobial agent to be used in aquaculture or animal husbandry.

## Figures and Tables

**Figure 1 ijms-23-00015-f001:**
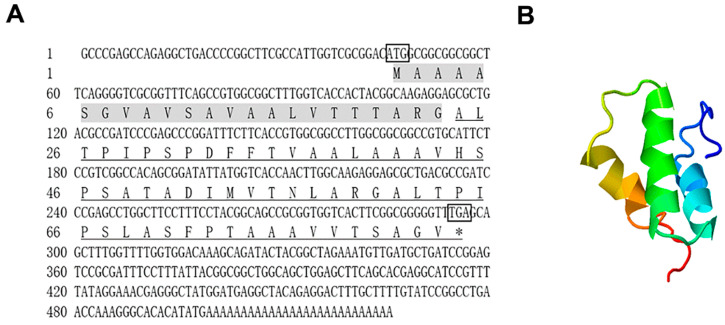
Bioinformatics analysis of Sparanegtin. The cDNA and deduced amino acid sequences of Sparanegtin: the boxed sequence represents the initiation codons; the boxed sequence and “*” represents the stop codons; the predicted signal peptide is shaded; underline regions indicate the mature peptide (**A**). The protein structure of Sparanegtin mature peptide was predicted by I-TASSER server (**B**).

**Figure 2 ijms-23-00015-f002:**
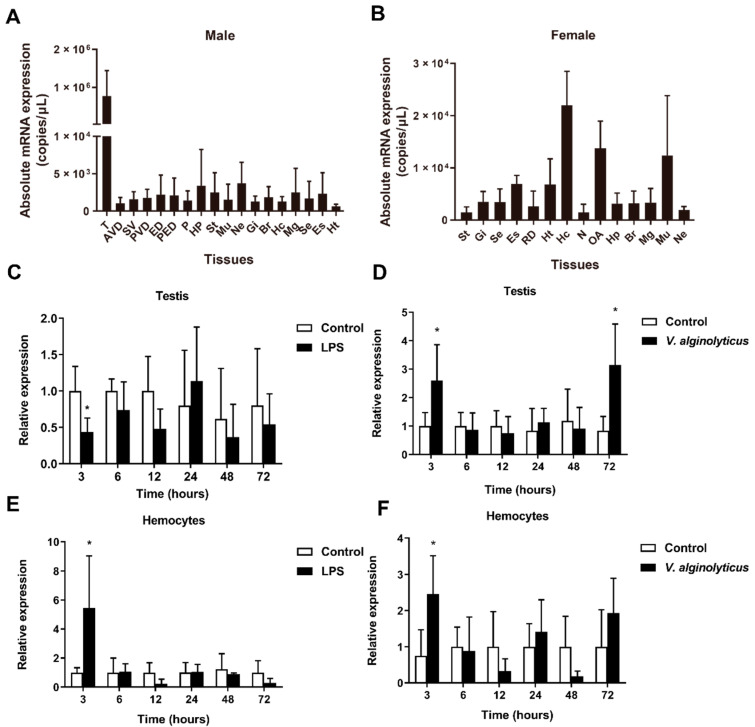
Gene expression profiles of *Sparanegtin* in *S. paramamosain*. The expression profile of *Sparanegtin* in adult male (**A**) and female (**B**) crabs under natural status was determined by absolute qPCR. Data are presented as mean ± standard deviation (SD). The relative expression of *Sparanegtin* in male testes after LPS (**C**) and *V. alginolyticus* (**D**) challenges, and in male hepatopancreas after LPS (**E**) and *V. alginolyticus* (**F**) challenges was examined. In panel (**C**–**F**), data are presented as mean ± SD. * *p* < 0.05, one-way analysis of variance (ANOVA) and Bonferroni post-test. Abbreviations: T, testis; AVD, anterior vas deferens; SV, seminal vesicle; PVD, posterior vas deferens; ED, ejaculatory duct; PED, posterior ejaculatory duct; P, penis; OA, ovary; N, spermathecae; RD, reproductive duct; Mu, muscle; Ne, thoracic ganglion; Gi, gills; Br, brain; Hc, hemocytes; Mg, midgut; Se, subcuticular epidermis; Es, eye stalk; Ht, heart; Hp, hepatopancreas; St, stomach.

**Figure 3 ijms-23-00015-f003:**
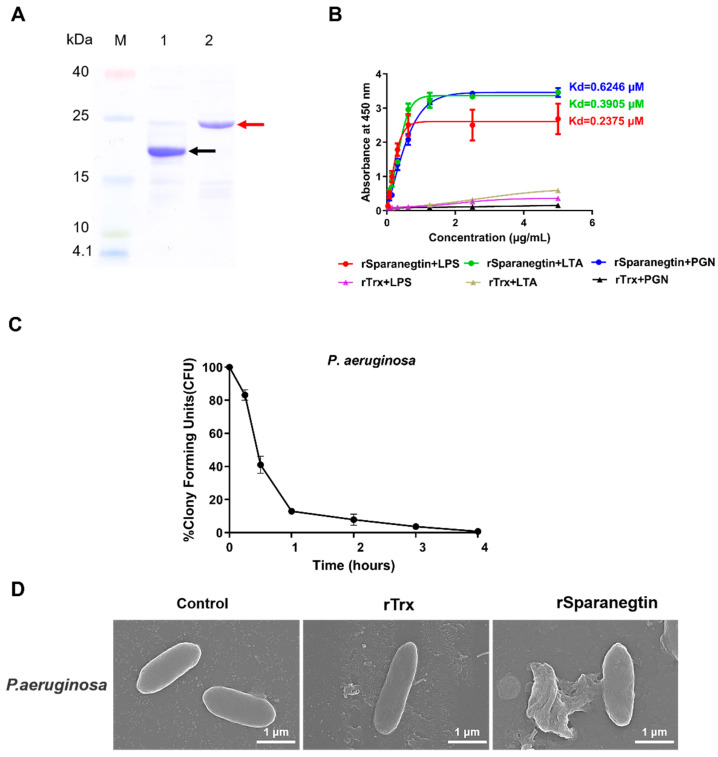
Binding activity and antibacterial mechanism of rSparanegtin. Expression and purification of recombinant Sparanegtin. Lane M: protein molecular standard; lane 1: purified rTrx; lane 2: purified rSparanegtin; the arrow indicates the size of the protein (**A**). Binding activity of rSparanegtin and rTrx to PAMPs (LTA for lipoteichoic acid, LPS for lipopolysaccharide, PGN for peptidoglycan) (**B**). Time-killing curves of *P. aeruginosa* treated with rSparanegtin (**C**). *P. aeruginosa* was suspended in culture media supplemented with PBS, rTrx, or rSparanegtin and observed by a scanning electron microscopy (SEM) (**D**).

**Figure 4 ijms-23-00015-f004:**
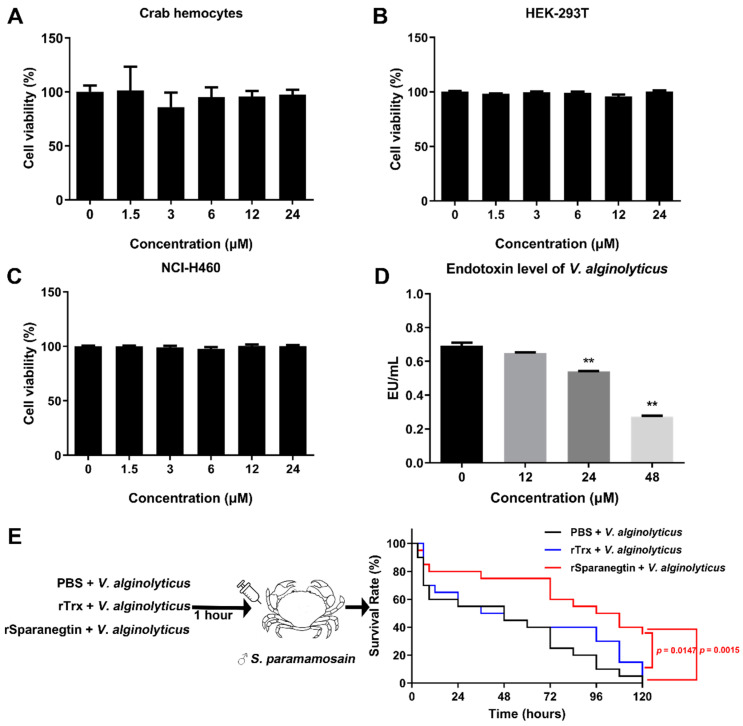
In vivo protective effect of rSparanegtin on *V. alginolyticus*-infected *S. paramamosain.* The cytotoxic effect of rSparanegtin on crab hemocytes (**A**), HEK-293T (**B**), and NCI-H460 (**C**) was determined by the MTS method; data are presented as mean ± standard deviation (SD) (*n* = 3). **: *p* < 0.01, one-way analysis of variance (ANOVA) and Dunnett post-test. Endotoxin level of *V. alginolyticus* after rSparanegtin treatment in vitro (**D**). In vivo protective effect of rSparanegti was evaluated (**E**). The rSparanegtin (20 µg/crab), rTrx (20 µg/crab), and PBS was incubated with *V. alginolyticus* (1 × 10^6^ CFU/crab) at room temperature for 60 min and then injected into the male crabs (*n* = 20 for each group). The survival curves were analyzed using the Kaplan–Meier Log rank test.

**Figure 5 ijms-23-00015-f005:**
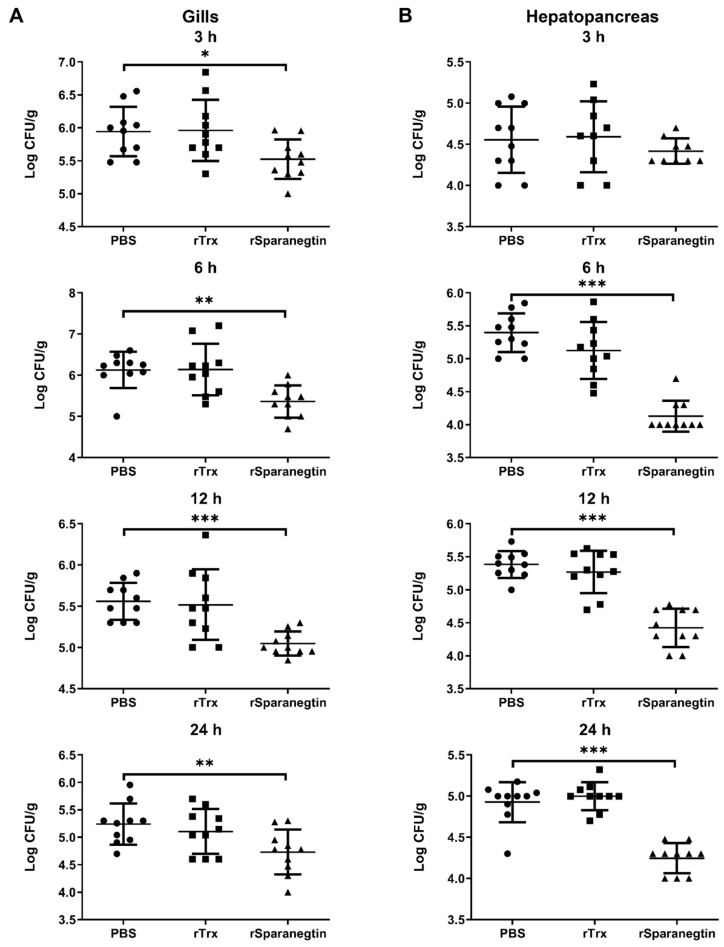
In vivo antimicrobial effect of rSparanegtin on *V. alginolyticus* growth in *S. paramamosain.* The rSparanegtin (20 µg/crab), rTrx (20 µg/crab), and PBS were incubated with *V. alginolyticus* (1 × 10^6^ CFU/crab) at room temperature for 60 min and then injected into the base of the right fourth leg of crabs. Infected crabs were dissected, and tissues including gills (**A**), midgut, and hepatopancreas (**B**) were collected at different time points (3, 6, 12, and 24 h). Homogenates were cultured onto marine broth 2216E plates. Colony numbers were normalized to tissue weight. Data represent the bacterial load in gills, midgut, and hepatopancreas. * *p* < 0.05, ** *p* < 0.01; *** *p* < 0.001.

**Figure 6 ijms-23-00015-f006:**
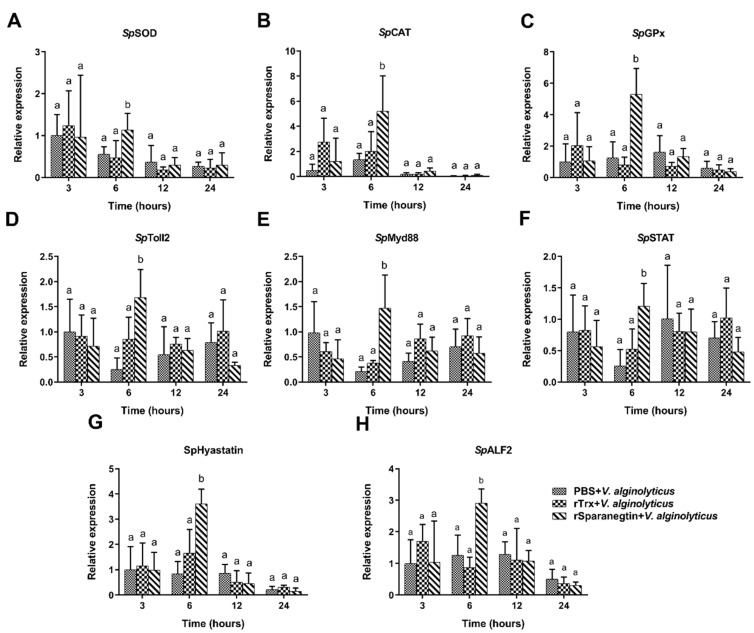
rSparanegtin effects on the *V. alginolyticus* infection-mediated immune gene expression profiles in *S. paramamosain.* Crabs were divided into PBS + *V. alginolyticus*, rTrx + *V. alginolyticus*, and rSparanegtin + *V. alginolyticus* groups. The expression levels of *Sp*SOD, *Sp*CAT, *Sp*GPx, *Sp*Toll2, *Sp*Myd88, *Sp*STAT, *Sp*ALF2, and SpHyastain (**A**–**H**) were evaluated using qPCR at 3, 6, 12, and 24 h post-injection. Each bar represents the means ± SD (*n* = 5). The same letters (a–b) indicate no significant difference between groups, and different letters indicate statistically significant differences between groups (*p* < 0.05) as calculated by one-way ANOVA followed by Tukey’s test. It was noted that only the means at each time point were compared for the denotation with the letters, whereas the means at different time points could not be compared with one another.

**Table 1 ijms-23-00015-t001:** Antimicrobial activity of rSparanegtin.

Microorganisms.	CGMCC No. ^a^	MIC (μM) ^b^	MBC (μM) ^b^
Gram-negative bacteria			
*Escherichia coli*	1.2389	24–48	24–48
*Pseudomonas aeruginosa*	1.2421	12–24	24–48
*Pseudomonas fluorescens*	1.1802	12–24	>48
*Aeromonas hydrophila*	1.2017	12–24	>48
*Shigella flexneri*	1.1868	12–24	>48
Gram-positive bacteria			
*Bacillus subtilis*	1.3358	24–48	>48
*Staphylococcus epidermidis*	1.4260	24–48	>48
*Staphylococcus aureus*	1.2465	12–24	>48
Fungi			
*Cryptococcus neoformans*	2.1563	24–48	>48
*Pichia pastoris (GS115)*	Invitrogen	24–48	>48

^a^ CGMCC No., China General Microbiological Culture Collection Number. ^b^ The MIC and MBC values are presented as the interval (A)–(B): (A) is the highest concentration tested with visible microbial growth, while (B) is the lowest concentration without visible microbial growth (*n* = 3).

**Table 2 ijms-23-00015-t002:** Primer sequences.

Primers	Sequence (5′−3′)
Sparanegtin-ORF-F	ATGGCGGCGGCGGCTTCAGG
Sparanegtin-ORF-R	TCAAACCCCCGCCGAAGTGA
Sparanegtin-5′-R1	CGGCTGCCGTAGGAAAGGAA
Sparanegtin-5′-R2	AATATCCGCTGTGGCCGACG
Sparanegtin-3′-F1	CGTCGGCCACAGCGGATATT
Sparanegtin-3′-F2	ACCAACTTGGCAAGAGGAGCG
Long primer	CTAATACGACTCACTATAGGGCAAGCAGTGGTATCAACGCAGAGT
Short primer	CTAATACGACTCACTATAGGGC
NUP	AAGCAGTGGTATCAACGCAGAGT
M13–47F	CGCCAGGGTTTTCCCAGTCACGAC
M13–48R	AGCGGATAACAATTTCACACAGGA
Sparanegtin-qPCR-F	TCCCCGGTTTCCCGACCCAG
Sparanegtin-qPCR-R	ACCAGGAGGCAGCACCGTCT
GAPDH-qPCR-F	CTCCACTGGTGCCGCTAAGGCTGTA
GAPDH-qPCR-R	CAAGTCAGGTCAACCACGGACACAT

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
