# Peer review of "A Novel Antimicrobial Peptide Sparanegtin Identified in Scylla paramamosain Showing Antimicrobial Activity and Immunoprotective Role In Vitro and Vivo"

_ijms, 2021, doi:10.3390/ijms23010015_

Round 1
Reviewer 1 Report
The manuscript entitled “A Novel Antimicrobial Peptide Sparanegtin identified in Scylla paramamosain showing anti- 2 microbial activity and immunoprotective role in vitro and vivo” investigated antimicrobial and immunoprotective properties of a newly identified AMP gene extracted from the mud crab Scylla paramamosain. The novel antimicrobial peptide, Sparanegtin, could be a good alternative to classic antibiotics. The presented results of high interest are within the scope of IJMS and provide useful information to define an adequate solution for a contemporary problem.The results are properly presented, and most of the methodological design is clear and appropriate. However, some remarks should be addressed before publication:
- The presented SEM image of the cells exposed to rSparanegtin is my concern. I am not convinced that what is observed on this figure is the bacterial cell. It may also be an artefact. How long were the cells incubated with PBS, rTrx 174 or rSparanegtin before imaging? How were the samples dehydrated before imaging?
- What is the relevance of the study? I recommend highlighting the novelty of the study. It is also necessary to make very clear the objective of the work to close the introduction section.
- In my opinion it would be useful to explain in the results section why the rTrx group was used.
- There is no conclusion section that should explain the main findings of the study.
Author Response
Response to Reviewer 1 Comments
Point 1: The presented SEM image of the cells exposed to rSparanegtin is my concern. I am not convinced that what is observed on this figure is the bacterial cell. It may also be an artefact. How long were the cells incubated with PBS, rTrx 174 or rSparanegtin before imaging? How were the samples dehydrated before imaging?
Response 1: Thanks for the reviewer’s comment, and we would like to give a brief explanation. The SEM image was obtained with the help of research assistant who is a staff at our University and specifically guide each user how to observe SEM. We have had a dozen of SEM images published in our previous papers (Shan, etc, 2016; Yang, etc, 2020; Zhu, etc, 2021) for other AMP studies. For our present study, the observed damage of bacterial cell membrane presented similar appearance, but the damage in the study looked much greater and the bacterial cell seemed all broken up and fragmented that might make the image not clear. To avoid the misunderstanding on SEM image, we have selected another one SEM image among lots of images taken by SEM in the study and substituted for the former figure in Figure 3 as the reviewer pointed out in the revised manuscript and also shown as below:
Figure 3. Binding activity, antibacterial mechanism of rSparanegtin. Expression and purification of recombinant Sparanegtin. Lane M: protein molecular standard; lane 1: purified rTrx; lane 2: purified rSparanegtin; the arrow indicates the size of the protein (A). Binding activity of rSparanegtin and rTrx to PAMPs (LTA for lipoteichoic acid, LPS for lipopolysaccharide, PGN for peptidoglycan) (B). Time-killing curves of P.aeruginosa treated with rSparanegtin (C). P. aeruginosa was suspended in culture media supplemented with PBS, rTrx or rSparanegtin, and observed by a scanning electron microscopy (SEM) (D).
References:
Shan, Z.; Zhu, K.; Peng, H.; Chen, B.; Liu, J.; Chen, F.; Ma, X.; Wang, S.; Qiao, K.; Wang, K. The New Antimicrobial Peptide SpHyastatin from the Mud Crab Scylla paramamosain with Multiple Antimicrobial Mechanisms and High Effect on Bacterial Infection. Front Microbiol 2016, 7, 1140, doi:10.3389/fmicb.2016.01140.
Yang, Y.; Chen, F.; Chen, H.Y.; Peng, H.; Hao, H.; Wang, K.J. A Novel Antimicrobial Peptide Scyreprocin From Mud Crab Scylla paramamosain Showing Potent Antifungal and Anti-biofilm Activity. Front Microbiol 2020, 11, 1589, doi:10.3389/fmicb.2020.01589.
Zhu, D.; Chen, F.; Chen, Y.C.; Peng, H.; Wang, K.J. The Long-Term Effect of a Nine Amino-Acid Antimicrobial Peptide AS-hepc3(48-56) Against Pseudomonas aeruginosa With No Detectable Resistance. Front Cell Infect Microbiol 2021, 11, 752637, doi:10.3389/fcimb.2021.752637.
For the questions “How long were the cells incubated with PBS, rTrx 174 or rSparanegtin before imaging? How were the samples dehydrated before imaging?”, we have described the methods of SEM much more in details and shown in line 468-475 in the revised manuscript as below:
“PBS, rTrx and rSparanegtin were separately added into each individual culture medium and incubated at a concentration of 48 µM for 30 min. The microbial cells were collected and fixed with pre-cooled 2.5% glutaraldehyde at 4°C for 2 h. The samples were then dehydrated with a graded series of ethanol(30%, 50%, 70%, 80%, 95% and 100%) and further dehydrated in a critical point dryer (EM CPD300, Leica, Germany) and gold coated. Finally, the change in morphology of the bacteria was observed by SEM (SUPRA 55 SAPPHIRE, Carl Zeiss, Germany).”
Point 2: What is the relevance of the study? I recommend highlighting the novelty of the study. It is also necessary to make very clear the objective of the work to close the introduction section.
Response 2: Thanks for the reviewer’s comment. We would like to give a brief explanation for the questions raised by reviewer. With the increasing emergence of antibiotic-resistance in almost bacteria, many countries have enacted regulations to limit the overuse of antibiotics in medicine or prohibit antibiotics to be as growth promoters in animal feeds in recent years. AMPs are considered as potential antibiotic substitutes, thus the purpose of our study is to explore new antimicrobial peptides from marine animals, especially from invertebrates. Following up, these newly identified AMPs will be developed to be as a bio-friendly and effective antimicrobial agent, which could be substituted for antibiotics to be used in animal husbandry or medicine. We have added the objective of the work in line 84-100 in the revised manuscript as below:
“In the study, based on the transcriptome database of S. paramamosain established by our laboratory, we identified an uncharacterized gene for the first time and named it Sparanegtin. The expression profiles of Sparanegtin in S. paramamosain with the challenge of LPS or V. alginolyticus were investigated. The recombinant product of Sparanegtin (rSparanegtin) in a prokaryotic expression system Escherichia coli was obtained. The antimicrobial activity assay, scanning electron microscopy (SEM) observation and microbial surface components binding assays were performed to analyze the antimicrobial features of rSparanegtin against various microorganisms in vitro. In addition, the effect of rSparanegtin in vivo was evaluated by detecting the bacterial clearance ability in the gills and hepatopancreas of S. paramamosain infected with V. alginolyticus, as well as any effect on expression patterns of some immune-related genes while in vivo administration of rSparanegtin. This study aims to characterize of the new AMP Sparanegtin, elucidate its immune protective effect and the underlying mechanism, thus developing a potential effective antimicrobial agent that could be substituted for antibiotics to be used in animal husbandry or medicine in future.”
Point 3: In my opinion it would be useful to explain in the results section why the rTrx group was used.
Response 3: Thanks for the reviewer’s comment. According to the reviewer comment, we have added a sentence in Line 156-158 in the revised manuscript and also shown as below:
“In order to evaluate whether the label protein Trx would have any effect on the following results, rTrx was selected as the control group.”
Point 4: There is no conclusion section that should explain the main findings of the study.
Response 4: Thanks for the reviewer’s comment. According to the reviewer comment, we have added the conclusion section in Line 538-548 in the revised manuscript and also shown as below:
“5. Conclusions
In summary, a new antimicrobial peptide named Sparanegtin was identified in S. paramamosain and its transcripts were specifically distributed in tissues and significantly expressed with bacterial challenge. rSparanegtin had antimicrobial activity and the antimicrobial mechanism involved initial damage to the outer membrane of bacteria, eventually resulting in the loss of cellular components and the complete collapse of the cell architecture. rSparanegtin showed no cytotoxicity and could reduce the V. alginolyticus endotoxin level in vitro. This AMP had an in vivo protective and immunomodulatory effect in S. paramamosain. that could reduce the bacterial load in tissues and enhance the survival rate of crabs challenged with V. alginolyticus. Taken together, Sparanegtin might be as a potential effective antimicrobial agent to be used in aquaculture or animal husbandry.”

Reviewer 2 Report
Could be used newer references.
Author Response
Response to Reviewer 2 Comments
Point 1: Could be used newer references.
Response 1: Thanks for the reviewer’s comment, and we have updated the reference list by some recently published literatures in the revised manuscript and also shown as below:
Previous manuscript:
- Dale, B.A.; Fredericks, L.P. Antimicrobial peptides in the oral environment: Expression and function in health and disease. Curr Issues Mol Biol 2005, 7, 119-133, doi:10.1016/j.engfracmech.2006.10.002.
- Molina-Cruz, A.; Dejong, R.J.; Charles, B.; Gupta, L.; Kumar, S.; Jaramillo-Gutierrez, G.; Barillas-Mury, C. Reactive oxygen species modulate anopheles gambiae immunity against bacteria and plasmodium. J Biol Chem 2008, 283, 3217-3223, doi:10.1074/jbc.M705873200.
- Pavlick, K.P.; Laroux, F.S.; Fuseler, J.; Wolf, R.E.; Gray, L.; Hoffman, J.; Grisham, M.B. Role of reactive metabolites of oxygen and nitrogen in inflammatory bowel disease. Free Radical Bio Med 2002, 33, 311-322, doi:10.1016/S0891-5849(02)00853-5.
Revised manuscript:
- Shabir, U.; Ali, S.; Magray, A.R.; Ganai, B.A.; Firdous, P.; Hassan, T.; Nazir, R. Fish antimicrobial peptides (AMP'S) as essential and promising molecular therapeutic agents: A review. Microb Pathogenesis 2018, 114, 50-56, doi:10.1016/j.micpath.2017.11.039.
- Lu, S.; Walters, G.; Parg, R.; Dutcher, J.R. Nanomechanical response of bacterial cells to cationic antimicrobial peptides. Soft Matter 2014, 10, 1806-1815, doi:10.1039/c3sm52801d.
- Dharmaraja, A.T. Role of Reactive Oxygen Species (ROS) in Therapeutics and Drug Resistance in Cancer and Bacteria. J Med Chem 2017, 60, 3221-3240, doi:10.1021/acs.jmedchem.6b01243.
- Kanzaki, H.; Wada, S.; Narimiya, T.; Yamaguchi, Y.; Katsumata, Y.; Itohiya, K.; Fukaya, S.; Miyamoto, Y.; Nakamura, Y. Pathways that Regulate ROS Scavenging Enzymes, and Their Role in Defense Against Tissue Destruction in Periodontitis. Front Physiol 2017, 8, doi:10.3389/fphys.2017.00351.

Reviewer 3 Report
The article is devoted to discovery of new AMP - so it is absolutely important to estimate toxicity on human cell lines - at least two, better on four or more (normal and cancer).
Not so crucial - but will be also important - all experiments were done with tagged protein - it will be good to reproduce at least part of results on peptide without tag (cleaved or chemically synthesized peptide)
It will be also good to detect this peptide by MS in natural samples.
Author Response
Response to Reviewer 3 Comments
Point 1: The article is devoted to discovery of new AMP - so it is absolutely important to estimate toxicity on human cell lines - at least two, better on four or more (normal and cancer).
Response 1: Thanks for the reviewer’s comment, and we agree with the reviewer that it is absolutely important to estimate toxicity on human cell lines. We have selected the human cell line HEK-293T and NCI-H460 to perform the toxicity assay and the results were also provided in this revised manuscript, showing that rSparanegtin has no toxic effect on both cells, as shown in Figure 4 in the revised manuscript and also shown as below:
Figure 4. In vivo protective effect of rSparanegtin on V. alginolyticus-infected S. paramamosain. Cytotoxic effect of rSparanegtin on crab hemocytes (A), HEK-293T (B) and NCI-H460 (C) was determined by MTS method, data are presented as mean ± standard deviation (SD) (n = 3). *P < 0.05, **: P< 0.01, one-way analysis of variance (ANOVA) and Dunnett post-test. Endotoxin level of V. alginolyticus after rSparanegtin treatment in vitro (D). In vivo protective effect of rSparanegtin was evaluated (E). The rSparanegtin (20 µg/crab), rTrx (20 µg/crab) and PBS was incubated with V. alginolyticus (1 × 10 6 CFU/crab) at room temperature for 60 min and then injected into the male crabs (n = 20 for each group). The survival curves were analyzed using the Kaplan-Meier Log rank test.
The relevant descriptions were also added as shown in Line183-185 and 477-490 in the revised manuscript:
“The cytotoxicity of rSparanegtin was analyzed using primarily cultured crab hemocytes, HEK-293T and NCI-H460. As shown in the Figure 4A, 4B and 4C, rSparanegtin showed no cytotoxicity.”
“The cytotoxicity of rSparanegtin was evaluated using primarily cultured crab hemocytes, HEK-293T and NCI-H460. The hemocytes of S. paramamosain were isolated as previously described (Deepika, etc, 2014). Briefly, the hemocytes were maintained in L-15 medium prepared in crab saline and supplemented with 5% fetal bovine serum, inoculated on a 96-well cell culture plate with approximately 104 cells well-1 and incubated overnight at 26°C. HEK-293T cells were maintained in Dulbecco’s Modified Eagle Medium supplemented with 10% fetal bovine serum and NCI-H460 cells were maintained in Roswell Park Memorial Institute 1640 supplemented with 10% fetal bovine serum. HEK-293T and NCI-H460 cells were inoculated on a 96-well cell culture plate and incubated at 37°C with 5% CO2 overnight. Finally, all the cells were incubated with culture medium supplemented with various concentrations of rSparanegtin (3, 6, 12, 24 and 48 μM, n=3). After 24 h of incubation, cell viability was assessed using a CellTiter 96 R ® AQueous Kit (Promega, America). The independent experiments were carried out three times.”
References:
Deepika, A.; Makesh, M.; Rajendran, K.V. Development of primary cell cultures from mud crab, Scylla serrata, and their potential as an in vitro model for the replication of white spot syndrome virus. In Vitro Cell Dev-An 2014, 50, 406-416, doi:10.1007/s11626-013-9718-x.
Point 2: Not so crucial - but will be also important - all experiments were done with tagged protein - it will be good to reproduce at least part of results on peptide without tag (cleaved or chemically synthesized peptide). It will be also good to detect this peptide by MS in natural samples.
Response 2: Thanks for the reviewer’s comments and we agree with the reviewer suggestion. We would like to give a brief explanation for that. During our present study, we ever tried several times to cleave the tag from the tagged protein as the reviewer mentioned, however, the collected quantity of protein without tag was much less than the tagged protein after additional treatments by enzymatic cleavage and second purification. Alternatively, we also tried to obtain a truncated peptide of Sparanegtin by chemical synthesis, but the synthesized peptide showed relatively weak antibacterial activity. Thus, we could not get the good results from the no-tag protein. Otherwise, we also agree with the reviewer suggestion that it will be good to detect this peptide by MS in natural samples. For this work, we have ever purified the natural protein Scygnonadin in our previous study in 2006 (Huang, etc, 2006), but we have not done the similar work in the study and we will further consider the reviewer’s instructive suggestion to do this work in our future study.
References:
Huang, W.S.; Wang, K.J.; Yang, M.; Cai, J.J.; Li, S.J.; Wang, G.Z. Purification and part characterization of a novel antibacterial protein Scygonadin, isolated from the seminal plasma of mud crab, Scylla serrata (Forskal, 1775). J Exp Mar Biol Ecol 2006, 339, 37-42, doi:10.1016/j.jembe.2006.06.029.

Round 2
Reviewer 1 Report
The manuscript has been sufficiently improved and can be published in IJMS.
Reviewer 3 Report
Accept